# Natural History of Hepatosplenic Schistosomiasis (HSS) Non–Cirrhotic Portal Hypertension (NCPH): Influence of Gastrointestinal Bleeding and Decompensation in Prognosis

**DOI:** 10.3390/tropicalmed8030145

**Published:** 2023-02-27

**Authors:** Zulane S. T. Veiga, Flávia F. Fernandes, Lívia Guimarães, Juliana Piedade, Gustavo Henrique S. Pereira

**Affiliations:** 1Gastroenterology and Hepatology Unit, Bonsucesso Federal Hospital, Ministry of Health, Rio de Janeiro 22640-100, Brazil; 2School of Medicine, Estácio de Sá University, Rio de Janeiro 25550-100, Brazil

**Keywords:** hepatosplenic schistosomiasis, liver disease decompensation, survival, non-cirrhotic portal hypertension

## Abstract

Background: Hepatosplenic schistosomiasis (HSS) is a peculiar form of non-cirrhotic portal hypertension (NCPH). Although HSS patients present normal hepatic function, some evolve signs of hepatocellular failure and features of decompensated cirrhosis. The natural history of HSS-NCPH is unknown. Methods: A retrospective study was conducted that evaluated patients who fulfilled clinical-laboratorial criteria for HSS. Results: A total of 105 patients were included. Eleven patients already presented with decompensated disease and had lower transplant-free survival at 5 years than those without (61% vs. 95%, *p* = 0.015). Among 94 patients without prior decompensation, the median follow-up was 62 months and 44% of them had varicose bleeding (two or more episodes in 27%). Twenty-one patients presented at least one episode of decompensation (10-year probability 38%). Upon multivariate analysis, varicose bleeding and higher bilirubin levels were associated with decompensation. The 10-year probability of survival was 87%. Development of decompensation and age were predictive of mortality. Conclusion: HSS is characterized by multiple episodes of GI bleeding, a high probability of decompensation and reduced survival at the end of the first decade. Decompensation is more common in patients with varicose esophageal bleeding and is associated with lower survival.

## 1. Introduction

Portal hypertension is a syndrome characterized by increased portal pressure and development of portosystemic collateral vessels, esophageal varices and splenomegaly, which is often due to cirrhosis of the liver [1]. Nevertheless, some patients develop portal hypertension in the absence of cirrhosis, a condition known as non-cirrhotic portal hypertension (NCPH) [2]. The most common cause of NCPH is schistosomiasis, but it may also be associated with the use of certain medications, exposure to toxins, genetic or immunological disorders, or it may be idiopathic (a condition previously called idiopathic noncirrhotic portal hypertension (INCPH), which was recently renamed as part of spectrum of porto-sinusoidal vascular disease) [2,3].

Schistosomiasis is a neglected tropical disease prevalent in low-to-middle income countries. Its transmission has been reported in 78 countries and the World Health Organization (WHO) estimates that more than 290 million individuals required preventive treatment against Schistosoma worldwide in 2018 [4]. It is a disease typically associated with poverty and a lack of adequate sanitation, which forces people into contact with unprotected, natural freshwater sources where transmission occurs [5]. The socioeconomic impact generated by this disease should not be underestimated since it affects productive young adults and school-age children, hampering growth and development. Despite advances in control and substantial decreases in morbidity and mortality, schistosomiasis continues to be an important public health issue in endemic countries [6]. Approximately 10% of infected individuals will develop the severe form of the disease characterized by portal hypertension and hepatic periportal fibrosis known as hepatosplenic schistosomiasis (HSS) [7,8].

Patients with HSS usually do not have impaired liver function since pathophysiology essentially involves inflammatory or fibrotic portal-obstructive vascular changes, leading to presinusoidal portal hypertension without significant repercussions either for the hepatic parenchyma or hepatic structural integrity [9]. Nevertheless, some patients have been described with signs of hepatocellular failure and features of decompensated cirrhosis (often described in individuals with previous episodes of varicose bleeding and even in need of liver transplantation) [10,11].

INCPH has largely been described in Oriental populations, and less frequently in Western countries. Characteristically, patients have well-preserved liver morphology and function and recurrent episodes of gastrointestinal bleeding related to portal hypertension. Other complications described in individuals with cirrhosis (such as ascites and hepatic encephalopathy) are often absent [2]. Some studies have demonstrated better prognosis after varicose gastrointestinal bleeding (VGIB) than cirrhotic patients, and good overall survival for this population, whilst others have not.

In contrast to INCPH, studies of the natural history of HSS-NCPH are lacking. Specifically, frequency of gastrointestinal bleeding, decompensated liver disease and survival are largely unknown. The aim of the present study is to assess the natural history of HSS-NCPH, with special emphasis on new-onset VGIB, frequency of decompensation and survival.

## 2. Materials and Methods

### 2.1. Study Population

The study group was composed of a retrospective cohort of patients followed in the Gastroenterology and Hepatology Unit of Bonsucesso General Hospital, a Brazilian Ministry of Health tertiary hospital in Rio de Janeiro, Brazil. Patients older than 18 years of age with an HSS diagnosis were included. The diagnosis of *Schistosoma mansoni* (*S. mansoni*) infection was based on the identification of characteristic periportal fibrosis by ultrasonography (US) in patients with an epidemiological history of contact with fresh water sources from endemic areas, as well as biochemical criteria (serological and/or coproscopic methods). Patients with viable eggs in feces or tissue biopsy (either liver or bowel) and those with a positive PCR for *S. mansoni* in stool were treated with Praziquantel.

The hepatosplenic form was defined by the presence of portal hypertension detected as esophageal and/or gastric varices with upper gastrointestinal endoscopy and/or splenomegaly or portosystemic collaterals on US/CT. Patients originally classified as having medium-sized esophageal varices, those with isolated gastric varices, as well as those with endoscopically treated varices at the beginning of follow-up were classified as large varices for study purposes. Primary prophylaxis with non-selective betablockers (NSBB) was instituted for patients with large esophageal varices without previous VGIB, and those intolerant or with contraindications to betablockers were treated by endoscopic variceal obliteration (either sclerotherapy or banding). Secondary prophylaxis was performed with a combination of NSBB and endoscopic treatment.

Exclusion criteria were as follows: moderate alcohol intake (up to 1 drink per day for women and up to 2 drinks per day for men), HCV, HBV, HIV infection or other chronic liver disease such as toxic, autoimmune or metabolic (including NAFLD). One hundred and forty-six patients treated both in the inpatient unit and in the outpatient clinics from 1984–2017 with portal hypertension, characteristic US signs of periportal fibrosis and positive epidemiology for *S. mansoni* were evaluated. The median time point of study recruitment was July 2005, and the majority of patients (92 patients, 88%) started follow-up at our institution from the year 2000.

### 2.2. Data Collection

At study inclusion, the following data were collected: socio-demographic variables such as sex and age, epidemiological variables, ultrasonographic and endoscopic data, previous varicose bleeding, and liver-related decompensation (ascites, hepatic encephalopathy and/or spontaneous bacterial peritonitis). Laboratory parameters (liver and kidney function as well as blood cell count) were recorded. Routine assessment included clinical examination, abdominal US, upper gastrointestinal endoscopy, blood cell count, liver and kidney function tests and thereafter on a regular basis.

Two analyses were performed: one for overall survival and the other for decompensation-free survival. For analysis of overall survival, patients were followed up until liver transplantation or death. For analysis of decompensation-free survival, patients were followed up until the first episode of liver disease decompensation.

### 2.3. Ethics Statement

The study was approved by an ethics committee. All patients gave signed, informed consent for participation in accordance with the principles of the Declaration of Helsinki (Revision of Edinburgh, 2000)

### 2.4. Definitions

Varicose gastrointestinal bleeding (VGIB)—clinically relevant digestive bleeding caused by the rupture of gastric esophageal varices in patients with portal hypertension [1].Ascites—abnormal accumulation of fluid inside the peritoneal cavity, identified by complementary exams (US/CT) or physical examination. Ascites can be graded from 1 to 3 according to the amount of fluid in the abdominal cavity [12].Spontaneous bacterial peritonitis (SBP)—bacterial infection of ascitic fluid without any intra-abdominal, surgically treatable source of infection [12].Hepatic encephalopathy (HE)—neuropsychiatric complication with a wide spectrum of symptoms that can affect patients with acute or chronic liver failure. HE is diagnosed and graded according to West-Haven criteria [13].*Anemia*, *leucopenia* and *thrombocytopenia* were defined by a serum hemoglobin value lower than 13 g/dL, a leucocyte count lower than 4.0 × 10^9^/L and a platelet count lower than <150 × 10^9^/L, respectively.

### 2.5. Statistical Analysis

Statistical analysis was carried out using the SPSS software, version 21.0 (SPSS Inc., Chicago, IL, USA). Categorical data are expressed as numbers (percentages) and continuous variables as mean ± standard deviation or median (interquartile range). Data were compared using the Student’s *t*-test for continuous variables and the chi-square test for categorical variables. Factors associated with the development of VGIB, decompensation and mortality were selected in univariate and multivariate analysis using binary logistic regression and Cox regression models (backward stepwise selection method), respectively. Survival curves were constructed using the Kaplan-Meier method and comparisons were performed using a log-rank test. A *p* value less than 0.05 was considered to be statistically significant. We assumed a prevalence of portal hypertension-related bleeding and liver-related decompensation of 20% each, based on median values derived from studies in patients with INCPH, as no values were available for HSS. For a level of confidence of 95% and a power of 90%, the minimal sample size would be 69 patients (OpenEpi software, version 3.01, available online at https://www.openepi.com/Menu/OE_Menu.htm, accessed on 18 February 2023).

## 3. Results

Among 146 patients evaluated, a total of 105 were eligible for the study according to Figure 1. Baseline characteristics of the entire cohort are shown in Table 1. Eleven patients already presented with decompensated liver disease at the beginning of follow-up. These patients had a higher frequency of previous varicose gastrointestinal bleeding and lower serum albumin and hemoglobin levels than those without. There was no difference in demographic or other laboratorial parameters related to portal hypertension and liver function, such as age, platelet count and bilirubin. Five-year transplant-free survival was considerably lower in these patients (61% vs. 95% for patients without decompensation, *p* = 0.015, log-rank test). Due to the strong effect of decompensation on survival, patients with previous decompensation at study inclusion were excluded from further analysis.

Among patients without previous decompensation, abnormal levels of either AST or ALT (>40 UI/L) were observed in 45% of patients, and both ALT and AST were simultaneously elevated in 20% of patients. Abnormal alkaline phosphatase levels (>2× ULN) were only observed in 8 patients. Frequency of arterial hypertension, diabetes mellitus, dyslipidemia and obesity (factors commonly associated with hepatic steatosis) were similar in patients with and without elevations in aminotransferases. Hypoalbuminemia (<35 g/L) and abnormal bilirubin levels (>1.0 g/dL) were observed in 15% and 60% of patients (bilirubin greater than 2 g/dL in 16%), respectively. Anemia, leucopenia and thrombocytopenia were observed in 50%, 52% and 81% of patients, respectively. Surgery for portal hypertension was combined with splenectomy in 12 patients, and 2 additional patients received isolated splenectomies. Median duration of follow-up was 62 months (IQR 21–140 months).

### 3.1. Variceal Bleeding

Forty-eight patients had at least one episode of VGIB, most prior to or at the beginning of the study (29 and 9 patients, respectively). Among those with prior VGIB, the index episode preceded study inclusion by a median of 24 months and, even though all were receiving NSBB, large esophageal varices were still present in 22 patients, whereas only 6 patients had eradicated varices (the remaining patient had small varices). In the remaining 9 patients, the first VGIB coincided with the beginning of follow-up, and all were receiving previous primary prophylaxis with NSSB.

Recurrence was common, being reported in 26 patients (2 or more episodes in 10 patients). There was no difference either in the frequency of recurrence or the number of patients with 2 or more episodes when comparing patients with VGIB prior to or at the beginning of the study (69% vs. 67% and 24% vs. 33%, *p* > 0.05 for both comparisons; chi-square test)

Among those without VGIB prior to the beginning of follow-up, 10 patients developed their first VGIB after a median of 24 months (IQR 4–73 months). At the beginning of the study, 9 of these patients had large esophageal varices and all were receiving betablockers. Recurrence of VGIB was observed in only one patient after 73 months. This frequency was much lower than that observed for patients with VGIB prior to the beginning of study follow-up (10% vs. 68%, *p* = 0.001; chi-square test).

In multivariate analysis, the presence of large esophageal varices, higher ALT and lower hemoglobin levels were predictors of first VGIB. Neither the use of betablockers, platelet count, nor the presence of portal vein thrombosis were associated with this complication (Table 2). The probability of variceal bleeding according to the presence of large esophageal varices adjusted by these 2 factors at baseline is shown in Figure 2.

### 3.2. Decompensation

Twenty-one patients developed decompensation of liver disease after a median of 52 (IQR 7-111) months. Ascites was the first liver-related decompensation in all of them. Nine patients developed further decompensations after ascites (SBP in 8, HE in 5, both in 4 patients). The 1-, 5- and 10-year probability of decompensation was 7%, 16% and 38%, respectively. VGIB, either prior to or after the beginning of follow-up, was more common among those patients who decompensated (80% vs. 48%, *p* = 0.01). The median time between VGIB and decompensation was 62 (IQR 2-117) months. Furthermore, there was no difference in the recurrence of VGIB between groups (56% vs. 43%, *p* = 0.63).

Table 3 shows the baseline factors associated with the development of decompensation. In multivariate analysis, only serum bilirubin and varicose GI bleeding were associated with decompensation. Figure 3 shows the probability of decompensation according to VGIB.

### 3.3. Survival

At the end of follow-up, 50 patients were alive, 7 were dead, 1 was submitted to a liver transplantation (after 54 months) and 38 patients were lost on follow-up (after a median of 51 (14–99) months]. The 1-, 5- and 10-year probability of survival was 99%, 92% and 87%, respectively.

Table 4 shows the baseline factors associated with survival. Age and development of decompensation, but not variceal bleeding, were independently associated with mortality. Figure 4 shows the survival rate according to the development of decompensation.

### 3.4. Other Complications

Two patients developed hepatocellular carcinoma, after 7 and 22 years of follow-up. Both had VGIB prior to study inclusion, but neither developed decompensations. Porto-pulmonary hypertension was not diagnosed in any patient.

## 4. Discussion

HSS-NCPH is a peculiar form of chronic liver disease with structural features and vascular changes that differ from cirrhosis [14]. Their pathophysiology involves inflammatory or fibrotic portal-obstructive vascular changes, leading to presinusoidal portal hypertension, usually with preservation of liver architecture and hepatic function [9]. Hence, it has always been considered a disease with a benign course even when compared to cirrhosis. This concept, albeit widespread, has largely been based on observation, as previous studies have nearly always focused on prognosis in patients submitted either to surgical treatment of portal hypertension or secondary prophylaxis for VGIB [15,16]. Our findings, however, do not support this notion.

VGIB was a common complication, observed in almost half of the patients, a frequency comparable to that in patients with INCPH [17,18]. The majority of them had already had the index episode before the beginning of follow-up. The clinical course of these individuals was marked by frequent relapse of hemorrhage despite secondary prophylaxis. However, none of them were submitted to other therapies for portal hypertension. One may wonder if the relatively preserved hepatic function over time and adequate recovery after these episodes did not prompt further evaluation. Nevertheless, studies have demonstrated that less invasive therapies for portal hypertension such as TIPS are safe and efficient in these individuals [19]. Finally, almost half of the patients only attended a specialized service more than 2 years after the onset of symptoms. This highlights difficulties in the care of this population, which frequently comprise migrants from endemic regions in Brazil (which is not the case for Rio de Janeiro) and who often live in rural or peripheral areas.

Though less frequent, VGIB was also observed during follow-up. The presence of large esophageal varices, similar to patients with INCPH and cirrhosis, was strongly associated with it. This occurred despite the frequent use of propranolol, a drug known to reduce portal pressure in these individuals [20]. Use of carvedilol, a drug with greater effect on portal pressure reduction in cirrhosis, has not been evaluated as a primary prophylaxis in these individuals. Studies on secondary prophylaxis in HSS with short-term follow-up and without measurement of portal pressure did not show superiority when compared to propranolol [21]. Early identification and alternative treatments for portal hypertension with good safety and efficacy profiles are still an unmet need in these individuals, who do not respond to pharmacological treatment.

Higher ALT also corelated with VGIB during follow-up. One possible explanation is that these individuals had a more intense granulomatous reaction around Schistosoma eggs (largely mediated by cytokines derived from type-2 CD4 + T-lymphocytes, especially IL-13) leading to more severe inflammatory, fibrotic and vascular changes and consequent elevated ALT levels. This would, finally, determine increased intrahepatic vascular resistance and portal pressure and increased risk of variceal rupture [22]. Unfortunately, liver biopsies and serum markers were not available to prove this hypothesis. It seems unlikely that this corresponds to the presence of concomitant liver disease, as patients with viral or alcoholic liver disease were excluded and components of metabolic syndrome were not more common in individuals with high ALT levels.

Finally, hemoglobin levels were also correlated with VGIB. In patients with cirrhosis, an inverse correlation between hemoglobin levels and portal pressure (as measured by hepatic venous pressure gradient) has been reported [23]. One possible explanation for this finding is that anemia aggravates hyperdynamic circulation in cirrhosis, consequently increasing portal pressure [24]. Hyperdynamic circulation has also been documented in HSS patients and this may explain the effect of hemoglobin levels on VGIB risk [25].

Liver-related decompensations were also common, both at the beginning of the study and at follow-up. In the present study, we observed that 20% of HSS patients presented at least one episode of decompensation during follow-up. Additionally, 10% already presented with decompensated liver disease at the beginning of follow-up, and this was associated with a worse outcome. Ascites was often the first and most common type of decompensation. Presence of VGIB and higher bilirubin levels were predictive factors for decompensation. It is well known that in HSS, intrahepatic portal vein obstruction and compensatory arterial hypertrophy render the hepatic parenchyma vulnerable to ischemic insult, consequent liver injury and gradual hepatocellular failure [14]. It should be noted that in HSS cases, total bilirubin elevation can be related to unconjugated bilirubin due to hemolysis. However, in decompensated HSS cases, high bilirubin levels may reflect end-stage liver disease as occurs in cirrhosis. Among patients with INCPH, the prevalence of decompensations at baseline varies between 9% and 34%, and between 5% and 19% at follow-up [17,18,26], which is somewhat similar to our findings. In INCPH populations, factors associated with increased risk of decompensation are currently unknown.

Development of decompensation, along with age, was associated with mortality. Development of ascites has been demonstrated as a predictive factor in patients with compensated cirrhosis [27], but also in INCPH [17,28]. Age has also been associated with increased rates of hospitalizations and deaths in patients with schistosomiasis in the State of Pernambuco, a highly endemic zone in the Northeast of Brazil [29]. The higher risk of death in older age groups may be explained by immunological and physiological aspects characteristic of age and the occurrence of more frequent chronic comorbidities in this population [30]. Only one patient was submitted to liver transplantation, in contrast to patients with INCPH, in whom this therapy has been used successfully [31]. We did not evaluate in detail the reasons why decompensated patients were not transplanted, but according to the present findings, we think this option should be considered for these patients.

Our study has some limitations. In spite of a detailed registry of patient clinical characteristics and follow-up, it has the inherent limitations of retrospective studies. Those patients included were treated over a large span of time and with different therapeutic options, a factor common among patient cohorts when investigating rare diseases. We also did not evaluate the incidence of portal vein thrombosis, a factor especially common in patients with HIV infection and associated with worse outcomes in some studies. Nevertheless, our frequency at baseline was much lower than reported for INCPH, which could make this complication unlikely to be of clinical relevance due to infrequency. Finally, the evolution of esophageal varices size was also not assessed. Instead, the size of varices at baseline was of prognostic significance, possibly reducing the relevance of follow-up assessments.

## 5. Conclusions

HSS is characterized by a relatively benign early course with a high probability of development of decompensation and reduced survival at the end of the first decade. Decompensation, either at diagnosis or during follow-up, is associated with lower survival. Variceal bleeding and worse liver function at baseline predicts decompensation. Patients with large esophageal varices are at an increased risk of bleeding.

## Figures and Tables

**Figure 1 tropicalmed-08-00145-f001:**
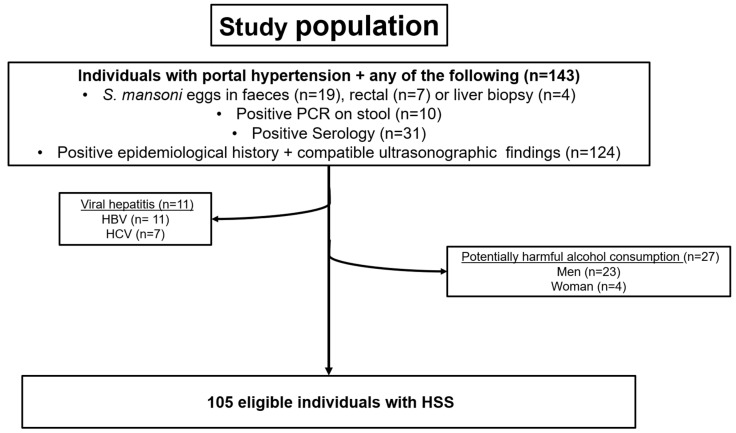
Study population. Frequencies in the first box (individuals with portal hypertension + any of the following) are not cumulative, as patients could have more than one of diagnostic criteria.

**Figure 2 tropicalmed-08-00145-f002:**
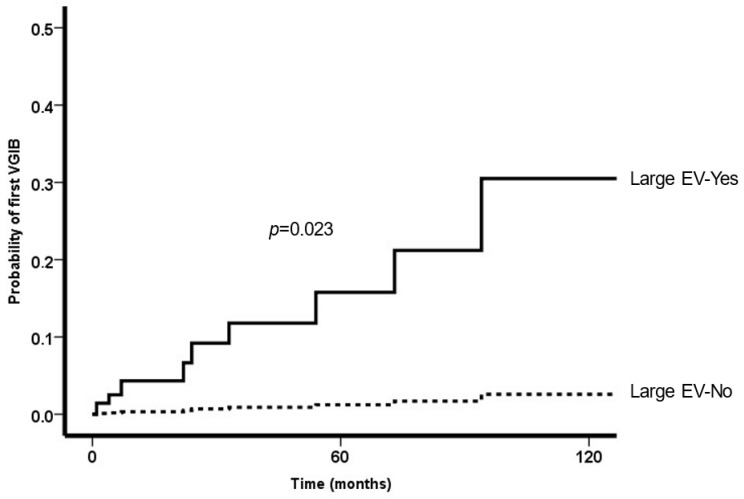
Probability of new-onset VGIB according to the presence of large esophageal varices (adjusted by ALT and Hb levels).

**Figure 3 tropicalmed-08-00145-f003:**
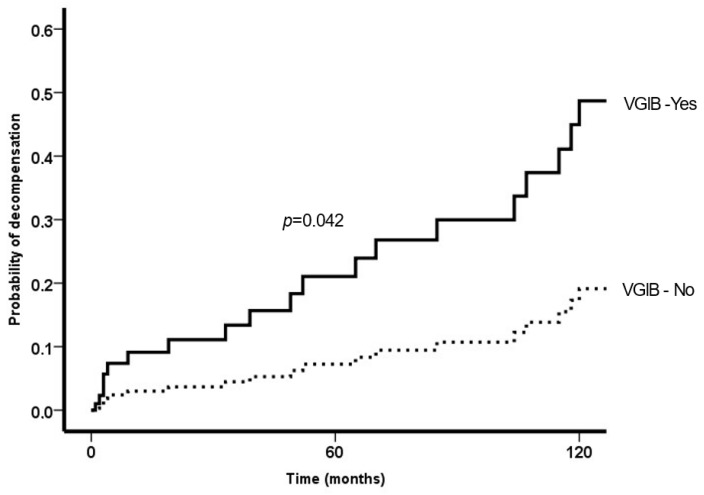
Probability of decompensation according to variceal bleeding (previous to or after follow-up), adjusted by bilirubin.

**Figure 4 tropicalmed-08-00145-f004:**
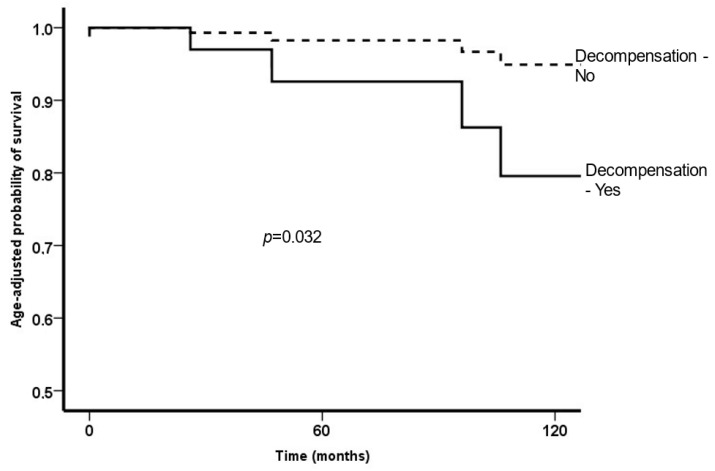
Age-adjusted survival according to development of decompensation.

**Table 1 tropicalmed-08-00145-t001:** Baseline characteristics of the study group, stratified by the presence of decompensations prior to the beginning of follow-up.

	Overall	Previous Decompensation		
		No (n = 94)	Yes (n = 11)	*p*-Value
Demographic data				
Age (years)	50 ± 13	50 ± 13	54 ± 16	0.40
Male sex	38 (36%)	35%	45%	0.52
Diabetes mellitus	26 (25%)	24%	27%	>0.99
Arterial hypertension	32 (31%)	30%	36%	0.73
Obesity	7 (7%)	7%	0%	0.34
Dyslipidemia	8 (8%)	9%	0%	0.59
Previous complications and procedures				
Variceal bleeding	48 (46%)	41%	73%	0.06
Surgery for portal hypertension	12 (11%)	12%	9%	0.79
Clinical characteristics				
Propranolol	86 (82%)	81%	90%	0.68
Mean arterial pressure (mmHg)	94 ± 15	94 ± 15	100 ± 13	0.36
Heart rate (bpm)	75 ± 11	75 ± 11	74 ± 8	0.87
Laboratorial characteristics				
ALT (U/L)	37 ± 21	37 ± 21	33 ± 14	0.52
AST (U/L)	42 ± 21	41 ± 20	48 ± 26	0.32
Alkaline phosphatase (U/L)	140 ± 86	142 ± 88	127 ± 48	0.62
GGT (U/L)	96 ± 83	94 ± 84	118 ± 80	0.37
Albumin (g/L)	39 ± 6	40 ± 5	34 ± 9	0.09
Bilirubin (mg/dL)	1.5 ± 1.6	1.5 ± 1.7	1.5 ± 1.9	0.93
INR	1.22 ± 0.23	1.2 ± 0.2	1.4 ± 0.3	0.20
Hemoglobin (g/dL)	12.0 ± 2.4	12.1 ± 2.4	10.7 ± 2.3	0.05
Leucocyte count (× 10^9^/L)	4.5 ± 2.3	4.4 ± 2.2	5.0 ± 2.5	0.38
Platelet count (× 10^9^/L)	104 ± 68	101 ± 66	134 ± 82	0.14
Creatinine (mg/dL)	0.9 ± 0.3	0.9 ± 0.3	0.9 ± 0.2	0.98
Sodium (mEq/L)	139 ± 4	139 ± 4	138 ± 2	0.10
Endoscopic and radiological characteristics				
Esophagogastric varices (either present or treated)	90 (86%)	85%	91%	0.60
Portal hypertensive gastropathy	30 (29%)	29%	27%	0.92
Portal vein thrombosis	10 (10%)	10%	9%	0.96

**Table 2 tropicalmed-08-00145-t002:** Factors associated with new-onset bleeding (n = 56).

	Univariate	Multivariate
Age (years)	1.03 (0.97–1.08)	
Male sex	0.58 (014–1.84)	
Diabetes mellitus	1.97 (0.51–7.52)	
Arterial hypertension	0.34 (0.07–1.59)	
Previous surgery	0.04 (0.001–183)	
Propranolol	32.5 (0.09–11.332)	
Mean arterial pressure (mmHg)	0.98 (0.93–1.02)	
Heart rate (bpm)	0.97 (0.91–1.01)	
ALT (U/L)	1.036 (1.013–1.058)	1.03 (1.01–1.06)
AST (U/L)	1.02 (0.998–1.046)	
Alkaline phosphatase (U/L)	1.004 (0.999–1.009)	
GGT (U/L)	1.003 (0.996–1.010)	
Albumin (g/L)	1.19 (0.34–4.18)	
Bilirubin (mg/dL)	0.57 (0.20–1.62)	
INR	4.55 (0.12–173)	
Hemoglobin (g/dL)	0.82 (0.65–1.03)	0.76 (0.59–0.97)
Leucocyte count (×10^9^/L)	0.89 (0.65–1.03)	
Platelet count (×10^9^/L)	0.99 (0.98–1.009)	
Creatinine (mg/dL)	2.37 (0.50–11.2)	
Esophageal varices	31.29 (0.08–12,029)	
Large esophageal varices	9.01 (1.14–71.21)	13.2 (1.45–133)
Portal hypertensive gastropathy	0.67 (0.14–3.13)	
Portal vein thrombosis	0.78 (0.09–6.71)	

**Table 3 tropicalmed-08-00145-t003:** Factors associated with decompensation at long-term (n = 94).

	Univariate	Multivariate
Age (years)	0.99 (0.96–1.04)	
Male sex	1.55 (0.61–3.92)	
Diabetes mellitus	1.28 (0.47–3.57)	
Arterial hypertension	2.41 (0.81–7.20)	
Previous complications		
Variceal bleeding (previous)	0.47 (0.19–1.15)	
Variceal bleeding (follow-up)	1.76 (0.59–5.26)	
Variceal bleeding (previous + after)	3.43 (1.14–10.25)	3.32 (1.01–10.0)
Surgery	2.32 (0.53–10.2)	
Clinical and Laboratorial		
Propranolol	25.6 (0.11–6056)	
Mean arterial pressure (mmHg)	0.98 (0.96–1.02)	
Heart rate (bpm)	0.98(0.95–1.03)	
ALT (U/L)	1.01 (0.99–1.03)	
AST (U/L)	0.99 (0.98–1.02)	
Alkaline phosphatase (U/L)	1.005 (1.001–1.009)	
GGT (U/L)	1.00 (0.99–1.006)	
Albumin (g/L)	0.86 (0.40–1.86)	
Bilirubin (mg/dL)	1.34 (1.12–1.60)	1.34 (1.12–1.60)
INR	2.07 (0.32–13.2)	
Hemoglobin (g/dL)	0.93 (0.78–1.10)	
Leucocyte count (×10^9^/L)	0.98 (0.83–1.17)	
Platelet count (×10^9^/L)	0.997 (0.99–1.005)	
Creatinine (mg/dL)	2.26 (0.57–8.91)	
Sodium (mEq/L)	0.92 (0.83–1.03)	
Endoscopy and Radiology		
Esophageal varices (any)	25.9 (0.13–5131)	
Large esophageal varices	2.89 (0.85–9.88)	
Portal hypertensive gastropathy	2.32 (0.96–5.79)	
Portal vein thrombosis	1.22 (0.36–4.16)	

**Table 4 tropicalmed-08-00145-t004:** Factors associated with mortality at long-term (n = 94).

	Univariate	Multivariate
Age (years)	1.08 (1.01–1.16)	1.08 (1.01–1.15)
Male sex	0.32 (0.08–1.30)	
Diabetes mellitus	1.41 (0.28–7.14)	
Arterial hypertension	0.58 (0.11–2.78)	
Previous complications		
Variceal bleeding		
At inclusion or before	1.86 (0.44–7.78)	
After inclusion	1.83 (0.37–9.10)	
Before or after inclusion	4.18 (0.51–34.1)	
Surgery	0.31 (0.03–34.2)	
Decompensations	5.89 (1.19–29.2)	6.00 (1.15–31.3)
Clinical and Laboratorial		
Propranolol	23.77 (0.001–622,492)	
Mean arterial pressure (mmHg)	0.97 (0.92–1.02)	
Heart rate (bpm)	0.98 (0.93–1.04)	
ALT (U/L)	0.9 (0.96–1.03)	
AST (U/L)	1.01 (0.98–1.04)	
Alkaline phosphatase (U/L)	1.002 (0.994–1.009)	
GGT (U/L)	1.004 (0.996–1.012)	
Albumin (g/L)	0.81 (0.25–2.60)	
Bilirubin (mg/dL)	1.15 (0.89–1.47)	
INR	1.89 (0.11–32.67)	
Hemoglobin (g/dL)	0.94 (0.72–1.25)	
Leucocyte count (×10^9^/L)	1.03 (0.72–1.25)	
Platelet count (×10^9^/L)	0.99 (0.98–1.01)	
Creatinine (mg/dL)	6.4 (0.45–91.3)	
Sodium (mEq/L)	0.79 (0.66–0.96)	
Endoscopy and Radiology		
Esophageal varices	25.18 (0.003–210,931)	
Large esophageal varices	2.86 (0.35–23.3)	
Portal hypertensive gastropathy	1.25 (0.77–2.02)	
Portal vein thrombosis	1.19 (0.38–9.58)	

## Data Availability

The data presented in this study are available on request from the corresponding author. The data are not publicly available due to privacy.

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
