# Peer review of "Natural History of Hepatosplenic Schistosomiasis (HSS) Non–Cirrhotic Portal Hypertension (NCPH): Influence of Gastrointestinal Bleeding and Decompensation in Prognosis"

_tropicalmed, 2023, doi:10.3390/tropicalmed8030145_

Round 1

Reviewer 1 Report

This is a very interesting study, providing some biochemical and specific pathological predictors for HSS-NCPH, which will be useful for clinicians, in particular in schistosomiasis endemic areas

Author Response

We are pleased to submit our revised manuscript tropicalmed-2197816, entitled “Natural history of hepatosplenic schistosomiasis (HSS) non–cirrhotic portal hypertension (NCPH): influence of gastrointestinal bleeding and decompensation in prognosis”, for your consideration as an original research article to Tropical Medicine and Infectious Disease. We would like to thank the reviewers for the insightful comments and suggestions for our manuscript; we believe the responses to these comments have resulted in a more relevant article.

We have implemented their comments and suggestions and wish to submit a second revised version of the manuscript for further consideration in the journal

Reviewer 2 Report

manuscript is well designed and well written.

Author Response

(The authors gave the same response as above.)

Reviewer 3 Report

Dear Authors,

Your manuscript entitled "Natural history of hepatosplenic schistosomiasis (HSS) non–cirrhotic portal hypertension (NCPH): influence of gastrointestinal bleeding and decompensation in prognosis." was reviewed,

In general the paper highlights a very important topic related to the parasitic protozoa "schistosoma sp.".

The paper is well prepared and well written in English, but it lacks a lot of modifications in its parts.

Kindly find below my remarks (Minor and Major) ones.

Minor Comments:

01- In the Abstract section, "Background, Methods, Results and Conclusion" terms should be in bold.

02- In the Introduction section, Line 39, authors are invited to add "against schistosoma" after the expression "required preventive treatment" so it would be as "required preventive treatment against schistosoma".

03- In some parts of the manuscript, authors are invited to put a space between the end of the sentence and the reference, example: Line 40 "2018 (4)" and Line 46 "countries (6)".

04- In the Materials and Methods section, Line 73, authors are invited to replace "schistosomiasis mansoni infection" by "schistosoma mansoni infection".

05- In the Materials and Methods section, Line 73, authors are invited to put (S. mansoni) after Schistosoma mansoni

06- In the Materials and Methods section, Line 134, authors are invited to replace "p value" by "P value".

07- In the whole manuscript, authors are invited to put figures' legends below the figures.

08- Figure 1 is not clear, authors are invited to put a clearer one.

09- Figure 1, last part, HES or HSS?

10- The Table 1's legend is poor in information, authors are invited to put another legend.

11- In the Results section, lines 144-145, authors are invited to explain from where did they got this result with the value.

12- Regarding all the tables, the form must be changed.

13- In the Results section, Line 174, authors are invited to explain from where did they got this result with the P value.

14- In the Results section, line 180, same remark as 12.

Major Comments:

01- In the Materials and Methods section, Line 93, the study population is made up of patients with inclusion criteria in the period between 1984 and 2017. I think that this period of time is very long, and medical tools and research is very different between these two years (33 years). This is a very weak point in the present study.

02- Authors are invited to move the section "Statistical Analysis" to the "Materials and Methods" section.

03- Authors are invited to calculate the power of the study with the sample size (104).

04- In the manuscript (Materials and Methods section) and in "Figure 1" authors talk about 104 eligible individuals to participate in this study. Authors are invited to explain why in the Table 1 the Overall number of participants is 105!!!.

05- In the Results section, Lines 164-165, authors are invited how they evaluated or count the episodes of VGIB prior or at the beginning of the study, since it is a retrospective study (started in 1984).

BR, 

Author Response

(The authors gave the same response as above.)

Reviewer 4 Report

The article entitled "Natural history of hepatosplenic schistosomiasis (HSS) non–cirrhotic portal hypertension (NCPH): influence of gastrointestinal bleeding and decompensation in prognosis" by Veiga et al. is a nice manuscript to read on. Hepatosplenic schistosomiasis (HSS) is a peculiar form of non–cirrhotic portal hypertension (NCPH), and authors have tried to decipher the natural history of HSS-NCPH, with special emphasis on new-onset VGIB, frequency of decompensation and survival. This was indeed interesting. Authors have successfully reached the goal of the study however, few things are to be checked before it can be accepted for publication.

1. the definition part should be removed from the text. It can be inserted as foot notes. This will essentially help to reduce the page numbers.

2. The font size for the figure 1 should be increased. They are not readable.

Rest of the paper is well written. Authors can slightly improve the language in the revised paper. Mostly I liked the honest approach of the authors  (limitation portion of this manuscript)

Author Response

(The authors gave the same response as above.)

Round 2

Reviewer 3 Report

Dear Authors your manuscript was re-reviewed.

I would like to thank you for all the effort you did, for all the modifications, the article is way better in its present form.

I have one Major comment that needs to be justified.

Authors are kindly invited to calculate the Power of the study population, they can use the following article as reference for this idea. (Prevalence, risk factors and seasonal variations of different Enteropathogens in Lebanese hospitalized children with acute gastroenteritis).

Best Regards,

Author Response

February 18th, 2023

Piyanuch Vibulcharoenkitja

Editor

Tropical Medicine and Infectious Disease

On behalf of my co-authors, I am resubmitting the attached manuscript, originally titled “Natural history of hepatosplenic schistosomiasis (HSS) non–cirrhotic portal hypertension (NCPH): influence of gastrointestinal bleeding and decompensation in prognosis” for reconsideration for publication in Tropical Medicine and Infectious Disease.

We thank you and the reviewers for your time and effort in reviewing the manuscript. The recommendations and advice have helped us to significantly enhance the quality of the manuscript.

We have revised our manuscript according to all of your comments, as explained below. We hope that our resubmission is now suitable for inclusion in Tropical Medicine and Infectious Disease and we look forward to hearing from you.

Sincerely,

Gustavo Pereira, M.D.; Ph.D.

Avenida Londres 616 (21041-030), 3rd floor, Bonsucesso, Rio de Janeiro (RJ).

E-mail: ghspereira@gmail.com

Tel: +55 21 39779893

Point-by-point reply to reviewer's comments: Reviewers’ comments in in regular font, authors’ responses in bold, significant changes to the manuscript in italics.

Review 3

I would like to thank you for all the effort you did, for all the modifications, the article is way better in its present form.

I have one Major comment that needs to be justified.

Authors are kindly invited to calculate the Power of the study population, they can use the following article as reference for this idea. (Prevalence, risk factors and seasonal variations of different Enteropathogens in Lebanese hospitalized children with acute gastroenteritis).

Best Regards,

Thank you for the comment. We followed the reference and made corresponding calculation. We included the following phrases on Statistical analysis section: “We assumed a prevalence of portal hypertension-related bleeding and liver-related decompensation of 20% each, based on median values derived from studies in patients with INCPH, as no values were available for HSS. For a level of confidence of 95% and a power of 90%, the minimal sample size would be 69 patients (OpenEpi software, version 3.01)”.
